# Paired Electrolysis Enabled Cyanation of Diaryl Diselenides with KSCN Leading to Aryl Selenocyanates

**DOI:** 10.3390/molecules28031397

**Published:** 2023-02-01

**Authors:** Wei-Bao He, Luo-Lin Tang, Jun Jiang, Xiao Li, Xinhua Xu, Tian-Bao Yang, Wei-Min He

**Affiliations:** 1College of Chemistry and Chemical Engineering, Hunan University, Changsha 410082, China; 2School of Chemistry and Chemical Engineering, University of South China, Hengyang 421001, China; 3National Engineering Research Center of Low-Carbon Processing and Utilization of Forest Biomass, Nanjing Forestry University, Nanjing 210037, China

**Keywords:** green chemistry, paired electrolysis, diaryl diselenides, aryl selenocyanates, cyanation

## Abstract

The first example of paired electrolysis-enabled cyanation of diaryl diselenides, with KSCN as the green cyanating agent, has been developed. A broad range of aryl selenocyanates can be efficiently synthesized under chemical-oxidant- and additive-free, energy-saving and mild conditions.

## 1. Introduction

Selenium-containing compounds have attracted the interest of synthetic organic chemists during the past few years, due to the fact that organic selenium compounds are widely found in pharmaceutical, agrochemical molecules, fluorescent molecular probes and promising biomaterials [1,2,3]. Among various selenium-containing compounds, aryl selenocyanates (ArSeCN) represent a fundamentally important class of heteroatom-containing compounds, and have gained a great deal of attention. They not only display a wide range of biological and pharmacological activities but also serve as highly versatile synthetic intermediates for the synthesis of various organic selenium compounds in organic synthesis [4,5,6,7,8,9]. As a consequence, considerable efforts have been devoted to constructing such molecules, and a series of synthetic protocols have been developed [10,11]. The traditional methods can be divided into two categories according to the reaction substrates (Figure 1a): (1) the reaction of phenylselenol or its derivative (PhSeH and PhSeCl) with a cyanating agent [12,13]; and (2) the reaction of various selenium-free aryl substrates with in-situ-generated selenocyanating agent (selenium powder/ trimethylsilyl cyanide, [14,15] selenium dioxide/malononitrile [16]) or the harmful potassium selenocyanate (KSeCN) [17,18,19,20,21]. Despite their achievements, the above-mentioned strategies generally require toxic and environmentally hazardous cyano sources; therefore, developing eco-friendly and more practical approaches for constructing such molecules from low-cost and non-toxic cyano sources are still highly desirable.

Thiocyanate salt is one of the most widely used inorganic sulfosalts, and has been considered as an attractive cyano source, given its non-toxicity, low cost and abundance [22]. Both thermal [23,24,25,26,27] and photocatalytic cyanation [28,29] with MSCN have been reported in recent years (Figure 1b). Very recently, Bhat et al. reported the visible-light-induced Rhodamine-6G-catalyzed cyanation of diphenyl diselenide and NaSCN by using pure oxygen as the oxidant and K_2_CO_3_ as the additive (Figure 1c) [30]. However, all the above-mentioned reactions require chemical oxidizing agents, photocatalysts and stoichiometric amounts of additives.

Organic electrochemical synthesis, as an environmentally friendly synthetic strategy, has emerged as a powerful tool in organic synthetic chemistry [31,32,33,34]. In comparison with the traditional methods that often proceed at strong oxidative or reductive conditions with elevated temperature or pressure in the presence of chemical oxidizing agents or reducing agents, electrochemical reactions are usually carried out under milder conditions by precisely varying the applied electrode potential. Therefore, electrosynthesis is usually compatible with highly functionalized substrates and has displayed great potential in both synthetic and bioconjugation chemistry [35,36,37,38,39,40,41]. In recent years, electrochemical cyanation with various cyanating agents (trimethylsilyl cyanide [42,43,44,45,46], cyanobenziodoxolone [47], AIBN [48], tosyl cyanide [49] and 4-cyanopyridine [50]) have been used to construct the cyanated products. In contradistinction, to the best of our knowledge, electrochemical cyanation with thiocyanate salts has never been reported. Paired electrosynthesis utilizes traceless electrons as the redox reagent, maximizing the congeniality through performing anodic oxidation and cathodic reduction simultaneously, and displays pronounced advances in energy efficiency and time efficiency [51,52]. In recent years, a series of paired electrolysis strategies have been developed for the sustainable synthesis of important fine chemicals [53,54,55,56,57,58]. Considering the importance of ArSeCN, and with our continued interest in developing green synthetic reactions [59,60,61,62,63,64,65,66], we herein report the paired electrolysis-enabled cyanation of diaryl diselenides and KSCN for synthesizing ArSeCN under chemical oxidant-free and mild conditions (Figure 1d). To the best of our knowledge, this is the first example of the paired electro-synthesis of cyano-containing compounds from KSCN under electrochemical conditions.

## 2. Results and Discussion

The electrochemical selenocyanation of PhSeSePh (**1a**) with KSCN (**2a**) was selected as a model reaction for probing various reaction parameters (Table 1). Using LiBF_4_ as the supporting electrolyte and MeCN as the sole solvent, the target phenyl selenocyanate **3a** could be obtained in 83% GC yield under 15 mA constant current in an undivided cell equipped with a graphite plate anode and platinum plate cathode (Table 1, entry 1). Changing the graphite plate anode to a platinum plate anode led to the formation of product 3a in 51% GC yield (entry 2). Performing the model reaction with graphite plate anode/copper plate anode, graphite plate anode/copper plate anode or graphite plate anode/copper plate anode resulted in the generation of phenyl selenocyanate 3a in 60–17% GC yields (entries 3–5). However, no desired product was observed for the template reaction using nickel plate or magnesium plate as the anode instead of platinum plate (entries 6 and 7). Only a trace amount of product **3a** was observed when LiBF_4_ was replaced by other lithium salts (entry 8). Conducting the reaction with tetrafluoroborate salts as the supporting electrolyte gave a trace amount of product **3a** (entry 9). The solvent effect imposed a key effect on the reaction efficiency. When the reaction was carried out in DMF, 27% GC yield of **3a** was obtained (entry 10). A trace amount of **3a** was detected by using EtOH, DCE, DMSO, THF or sole acetone as the solvent (entry 11). No reaction was observed under 10 mA constant current electrolysis conditions (entry 12). Increasing the constant current from 15 mA to 20 mA gave a 65% yield of **3a** (entry 13). The omission of LiBF_4_ led to no transformation (entry 14).

Having established the optimized conditions (Table 1, entry 1), the substrate applicability of this electrochemical cyanation was investigated (Figure 2). Various substituted diphenyl diselenides possessing electron-neutral (H and Ph), electron-donating (Me, *^t^*Bu, OMe, OBn, OCF_3_ and SMe) or electron-withdrawing groups (F, Cl, Br, CF_3_ and CO_2_Me) at the *para*-position of the phenyl ring successfully entered this process, leading to the desired phenyl selenocyanates (**3a–3m**) in good yields. Both *ortho*- and *meta*-substituted diphenyl diselenides were suitable for this transformation, delivering the corresponding products (**3n**–**3q**) in 62–73% yields. The sterically bulky 1,2-dimesityldiselane participated in this reaction smoothly and gave the product **3r** in 74% yield. Diselenides substrates with fused aromatic rings, including naphthalene (**1s** and **1t**), 2,3-dihydrobenzo[b][1,4]dioxine (**1u**), quinoline (**1v**), indole (**1w**) and benzofuran (**1x**) could also be transformed into the desired products (**3s**–**3x**) in good yields.

To acquire insight into the mechanism of the cyanation reaction, a series of mechanistic studies were carried out. First, the radical scavenger experiment was conducted. The electrocatalytic reaction of diphenyl diselenide and KSeCN was fully suppressed in the presence of 2,2,6,6-tetramethylpiperidine-*N*-oxyl (TEMPO) and butylated hydroxytoluene (BHT) (Figure 3a). In addition, the diphenylethylene-CN adduct (**4a**) was detected by GC−MS. These results suggested that the cyano radical was the critical radical intermediate in the present transformation (Figure 3b). The phenylselanethiol (**4b**) was detected in the acidified reaction mixture by GC−MS, indicating that the phenylselanethiol anion was generated in the current cyanation reaction (Figure 4c). The power on/off experimental results indicated that continuous electric current was necessary to drive this cyanation reaction, thus excluding the radical chain mechanism.

To further understand the mechanism of the electrocatalytic process, cyclic voltammetry experiments (CV) were conducted (Figure 1, for details, see Appendix A). The CV curve of diphenyl diselenide presented an onset potential at 1.05 V and an oxidative potential at 1.54 V. Both lower onset potential (0.53 V) and lower oxidative potential (0.98 V) of KSCN were obtained. The mixture of diphenyl diselenide and KSCN had the same onset potential and the same oxidative potential as that of KSCN. These results suggested that the oxidation of KSCN might occur preferentially at the surface of the anode.

On the basis of the above-mentioned results and the previous literature, [27,45,61] a plausible mechanism for the electrochemical cyanation was depicted in Figure 4. Firstly, KSCN (2) underwent one-electron oxidation at the surface of the graphite anode to generate a thiocyanate radical (SCN^−^), which attacked the Se atom of PhSeSePh (1) to form the benzenesulfinoselenoyl cyanide intermediate (A). The unstable intermediate A easily underwent bond cleavage to produce the desired product PhSeCN (3) and phenylselanethiol radical B, followed by one-electron reduction to form phenylselanethiol anion C.

## 3. Materials and Methods

### 3.1. General Considerations

Unless otherwise noted, all reagents were obtained from commercial suppliers and used without further purification. The instrument used for electrolysis was a dual display potentiostat (DJS-292B) (made in China). Graphite electrode was 15 mm × 10 mm × 2 mm and platinum electrode was 15 mm × 10 mm × 0.1 mm. The instrument used for cyclic voltammetry was a CHI 660E potentiostat, and the conditions were as follows: a glassy carbon disk working electrode (diameter, 2 mm), Pt disk and Ag/AgCl (0.1 M in CH_3_CN) as counter and reference electrode. Thin-layer chromatography (TLC) employed glass 0.25 mm silica gel plates. Flash chromatography columns were packed with 200–300 mesh silica gel. ^1^H NMR spectra were recorded at 500 MHz and ^13^C NMR spectra were recorded at 126 MHz by using a Bruker Avance 500 spectrometer. Chemical shifts were calibrated using residual undertreated solvent as an internal reference (^1^H NMR: CDCl_3_ 7.26 ppm, ^13^C NMR: CDCl_3_ 77.0 ppm), the chemical shifts (δ) were expressed in ppm and J values were given in Hz. HRMS were performed on a spectrometer operating on ESI-TOF.

### 3.2. Typical Procedure for the Synthesis of **3**

In an undivided flask (10 mL) equipped with a stir bar, diphenyl diselenide (0.2 mmol), KSCN (0.5 mmol), LiBF_4_ (0.1 mmol) and MeCN (6 mL) were added. The flask was equipped with graphite (15 mm × 10 mm × 2 mm) as the anode and a platinum electrode (15 mm × 10 mm × 0.1 mm) as the cathode. The reaction mixture was stirred and electrolyzed at a constant current of 15 mA under room temperature for 9 h. After completion, the solvent was concentrated under reduced pressure, and the pure products **3** were obtained by flash chromatography on silica gel.

## 4. Conclusions

In conclusion, we have developed the first example of electrochemical cyanation of diaryl diselenides with KSCN as a cyanating agent through a paired one-electron oxidation and reduction processes. The importance and advantages of the present strategy is three-fold: (a) this work not only revealed a novel route for producing various aryl selenocyanates but also offered mechanistic insights into this reaction, which may suggest new processes for the construction of Se-CN bonds. (b) The present procedure was free of transition metals, chemical oxidants and poisonous reagents, making it more environmentally friendly. (c) The reaction proceeded with good functional-group compatibility, which should contribute to the practical synthesis of complex molecules. Given the natural abundance of inexpensive starting materials, good functional-group tolerance, good yields and simple operation procedures, the developed reaction is expected to be widely applied in synthetic chemistry and the pharmaceutical industry.

## Data Availability

Not applicable.

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
