# Peer review of "Paired Electrolysis Enabled Cyanation of Diaryl Diselenides with KSCN Leading to Aryl Selenocyanates"

_molecules, 2023, doi:10.3390/molecules28031397_

Round 1

Reviewer 1 Report

Aryl selenocyanates (ArSeCN) represent a fundamentally important class of cyano-containing compounds, and have gained increasing attention. In this manuscript, He, Xu and coworkers disclosed the first example of electrochemical cyanation of diaryl diselenides with KSCN as a cyanating agent through a paired one-electron oxidation and reduction processes, delivering a broad range of aryl selenocyanates efficiently under chemical oxidant-, additive-free, energy-saving and mild conditions. Furthermore, this work offered mechanistic insights into this reaction, which may suggest new processes for the construction of Se-CN bonds.

Therefore, this reviewer recommended the acceptance after minor revisions.

1)       Several recently literatures to synthesize selenides should be cited, such as Org. Lett. 2022, 24, 2175; Org. Lett. 201921, 3653; CCS Chem. 20213, 1423.

2)       In figure, the authors should use three huge different colors to draw these curves.

3)       In mechanism, the addition of 2 to cyano group sound not reasonable. It is possible NCS radical attacked Se in PhSeSePh to form PhSe radical. Then Se radical attacked CN group?

Reviewer 2 Report

The manuscript molecules-2151722 entitled as “Paired Electrolysis Enabled Cyanation of Diaryl Diselenides with KSCN Leading to Aryl Selenocyanates” was reviewed carefully. It seems interesting. The article is well written. The information and data are complete to prove the author's claim. I suggest that the article be published. So, it needs minor revision, due to some of the specific comments which are listed below:

1.     Regarding figure 1, no information is given in the manuscript!

2.     Scheme 3, equation a: It is not clear, please write more clearly.

3.     Line 103: (Scheme 3c) should change to (Scheme 4c)

4.     Scheme 3, equation b: 3aa should change to 3a

Author Response

Thanks for the kind comment and valuable suggestion.

  1. Regarding figure 1, no information is given in the manuscript!

Response: Sorry for our carelessness. We have provided these information.

“To further understand the mechanism of the electrocatalytic process, cyclic volt-ammetry experiments (CV) were conducted (Figure 1). The CV curve of diphenyl diselenide presented an onset potential at 1.05 V and an oxidative potential at 1.54 V. Both lower onset potential (0.53 V) and lower oxidative potential (0.98 V) of KSCN was obtained. The mixture of diphenyl diselenide and KSCN have a same onset potential and a same oxidative potential as that of KSCN. These results suggested that the oxidation of KSCN might occur preferentially at the surface of anode.”

  1. Scheme 3, equation a: It is not clear, please write more clearly.

Response: We have re-written Scheme 3a.

  1. Line 103: (Scheme 3c) should change to (Scheme 4c)

Response: We have revised the expression.

“The phenylselanethiol (4b) was detected in the acidified reaction mixture by GC-MS, indicating that the phenylselanethiol anion was generated in this cyanation reaciton (Scheme 4c)”

  1. Scheme 3, equation b: 3aa should change to 3a

Response: We have corrected this spelling mistake in Scheme 3b.
